# Controlled Symmetry with Woods-Saxon Stochastic Resonance Enabled Weak Fault Detection

**DOI:** 10.3390/s23115062

**Published:** 2023-05-25

**Authors:** Jian Liu, Jiaqi Guo, Bing Hu, Qiqing Zhai, Can Tang, Wanjia Zhang

**Affiliations:** 1College of Information Engineering, Nanjing University of Finance and Economics, Nanjing 210023, China; liujian@nufe.edu.cn (J.L.); guojiaqi.orange@foxmail.com (J.G.);; 2College of Modern Posts, Nanjing University of Posts and Telecommunications, Nanjing 210003, China; 3College of Science, Nanjing University of Posts and Telecommunications, Nanjing 210003, China

**Keywords:** weak fault detection, controlled symmetry with Woods-Saxon stochastic resonance (CSwWSSR), particle swarm optimization (PSO), signal-to-noise ratio (SNR)

## Abstract

Weak fault detection with stochastic resonance (SR) is distinct from conventional approaches in that it is a nonlinear optimal signal processing to transfer noise into the signal, resulting in a higher output SNR. Owing to this special characteristic of SR, this study develops a controlled symmetry with Woods-Saxon stochastic resonance (CSwWSSR) model based on the Woods-Saxon stochastic resonance (WSSR), where each parameter of the model may be modified to vary the potential structure. Then, the potential structure of the model is investigated in this paper, along with the mathematical analysis and experimental comparison to clarify the effect of each parameter on it. The CSwWSSR is a tri-stable stochastic resonance, but differs from others in that each of its three potential wells is controlled by different parameters. Moreover, the particle swarm optimization (PSO), which can quickly find the ideal parameter matching, is introduced to attain the optimal parameters of the CSwWSSR model. Fault diagnosis of simulation signals and bearings was carried out to confirm the viability of the proposed CSwWSSR model, and the results revealed that the CSwWSSR model is superior to its constituent models.

## 1. Introduction

As the industrialization has found its way into society, the safety and reliability of engineering systems and mechanical equipment play a critical role in the production process, which is an essential subject involving technology and people’s livelihoods. It is common knowledge that mechanical failures occur from time to time with the increase in the age of the equipment or the imperfection of the technology in the use of mechanical equipment. Therefore, it is advantageous to find equipment problems as soon as possible because doing so can minimize financial damage and potentially save lives. Yet, in real-world applications, mechanical equipment operates in the presence of very strong background noise, which is manifested as a high frequency of noise overwhelming the presence of the effective signal in the time and frequency domain diagrams. Therefore, the development of novel methods that can identify and extract useful signals under the interference of substantial background noise has become an increasingly important topic in the field of weak fault detection [1,2].

In general, it is believed that the presence of noise will obstruct the detection of effective signals. The previous approaches to weak fault detection, such as filter technique, empirical mode decomposition, singular value decomposition, and wavelet transform, all use a noise suppression viewpoint [3,4,5]. These signal processing methods based on noise cancellation, however, are inevitably interfering with useful signals that may even lead to compromised detection performance. In their investigation of paleometeorological variability, Benzi et al. initially put forth the stochastic resonance (SR) theory [6,7,8]. Researchers began to notice the phenomenon of noise-enhanced weak periodic excitation in ring lasers in the following years, and as a result, SR started to spark a research boom in a wide range of fields. For the first time, Hu et al. utilized SR to mechanical problem diagnostics in [9]. The traditional view of noise suppression has been challenged by the development of SR theory, and the notion of turning noise into valuable signal has captured the interest of many academics and researchers [10].

Over the past two decades, SR has been used in many different fields, including meteorology [11], pharmacology [12], nanomechanics [13,14,15], and medicine [16]. SR-aided signal processing is particularly useful for wireless communication [17], mechanical fault diagnosis [18], underwater target identification [19], image enhancement [20], and other applications. SR was found in [21] to improve the performance of the system, especially for the case of neural networks whose hidden layer consists of a threshold function. Nishiguchi and Fujiwara demonstrated that weak fault characteristic signals drowned in noise can be detected by a classical bistable stochastic resonance (CBSR) method performed by nano field effect transistors [22]. Li et al. proposed a noise-controlled second-order CBSR system method based on wavelet transform, and successfully implemented this method in wind turbine vibration signals [23]. Liu et al. applied SR to a lake eutrophic ecosystem and discussed in detail the effect of noise, time delay on the state transition of the lake in [24]. To overcome the restriction that CBSR can only be applied to periodic signals, Liu et al. concentrated on applying CBSR to binary pulse amplitude modulation (BPAM) signals [25]. He et al. proposed a detection method for non-collaborative SR for cognitive radio networks, which alleviates the signal-to-noise ratio (SNR) wall and the corresponding noise uncertainty problem of conventional energy detectors, especially in the case of low SNRs [26].

For the sake of practical engineering needs, time-delayed SR systems have attracted the attention of many scholars. In [27], Shi et al. proposed a time-delayed feedback tristable system and analyzed the effects of the time-delayed terms on the signal-to-noise ratio (SNR) and the probability density function (PDF), where they concluded that the time-delay can influence the SR phenomenon. Lu et al. [28] proposed a nonstationary weak signal detection strategy based on a time-delayed feedback SR model and demonstrated that it is applicable to the detection of strong nonlinear nonstationary signals. Mei et al. studied SR with time delay, correlated noise, and periodic signals using SNR theory to determine how time delay affected SR [29]. The SR problem in time-delayed bistable systems subject to Gaussian white noise was covered by He et al. in [30], along with an analysis of the impact of each parameter on the average first crossing time, the Shannon entropy, and the SNR.

The CBSR model has received the most attention among the several theoretical representations of SR. However, CBSR is only effective for small-amplitude signal identification due to the limitations of its own potential function. The frequent output saturation that CBSR encounters, which limits its ability to enhance performance in weak signal detection, is clearly described in [31,32,33,34]. In an effort to address this weakness of CBSR, Qiao et al. presented an adaptive unsaturated bistable stochastic resonance approach that, by adopting segmented potential walls with a fixed potential structure in [35] rather than the steep potential walls.

With the innovative development of SR, the model is not only limited to the CBSR, but also the multistable and monostable SR is gradually researched by scholars. Lu et al. investigated a tri-stable stochastic resonance (TSR) system and adaptively adjusted the parameters using particle swarm optimization (PSO) to obtain the best SNR and spectral output in [36]. Through simulation and actual fault signal detection, Lu et al. in [37] demonstrated the superiority of the proposed Woods-Saxon stochastic resonance (WSSR) method over the CBSR. Zhang et al. proposed a combined hybrid TSR system by combining WSSR and CBSR with simulations to demonstrate its effectiveness [38]. However, the TSR suggested in [38] is unable to controllably change the potential structure since the potential well depth and width of the CBSR change simultaneously with a single parameter. The two TSR models mentioned above have a significant impact on the potential structure when a parameter is changed [36,38]. Hence, building a novel controlled potential structure with an optimal control strategy to catch weak signals remains a difficult problem.

In this paper, we propose a novel SR model called controlled symmetry with Woods-Saxon stochastic resonance (CSwWSSR). Since the parameters of the WSSR model control the potential wells singly and explicitly, we construct a controlled TSR based on the WSSR with three potential wells controlled by different parameters. In order to establish the optimal control strategy with the highest SNR, we then confirm the ideal parameters for the CSwWSSR model in various noise environments by utilizing PSO in the scenarios of controlled potential well depth and controlled potential well width. The feasibility and superiority of CSwWSSR for practical engineering applications are further verified by comparing the results with the original signal and its constituent model through the detection experiments of simulation signal and bearing.

The rest of this essay is structured as follows. The CSwWSSR model and its components are introduced in Section 2, which is then followed by a brief explanation of the procedure for optimizing the CSwWSSR parameters. Section 3 confirms the usefulness of the proposed CSwWSSR-based weak fault detection in simulation signals, and presents an analysis of its performance under various noise intensities and various fault characteristic frequencies. In Section 4, We demonstrate the efficiency of the CSwWSSR model by applying it to the bearing experiments. Finally, we conclude this paper in Section 5.

## 2. The Proposed CSwWSSR

### 2.1. Introduction of CSwWSSR

Most of the studies on SR primarily investigated the classical bistable SR model based on the four-time reflection-symmetric potentials. In these models, a change in a single parameter may induce a large change in the shape of the potential function. However, the shape of Woods-Saxon potential illustrated in Equation (Equation 1) can be adjusted separately or combinedly with the parameters *H*, *W*, and *a*. The potential well of WSSR is depicted in Figure 1 with steep potential well walls and a smooth bottom. Comparing the four curves in Figure 1, *H* is to control the depth of the potential well, *W* is to control the width of the potential well, and *a* is to determine the steepness of the potential well.
(1)$$U_{1}\left(x\right)=-\frac{H}{1+exp(\frac{\left|x\right|-W}{a})}$$

Based on the previous analysis of WSSR, we propose a novel stochastic resonance model for weak fault detection, named CSwWSSR, whose potential function is expressed as
(2)$$U\left(x\right)=U_{1}\left(x\right)+U_{2}\left(x\right)=-\frac{H}{1+exp\left(\frac{\left|x\right|-W}{a}\right)}-\frac{1}{2}A\omega_{0}^{2}x^{2}+\frac{1}{4}B\beta x^{4}$$
where $U_{1}\left(x\right)$ indicates the potential function of WSSR as Equation (Equation 1), and $U_{2}\left(x\right)$ is the potential function of controlled symmetric stochastic resonance (CSSR), which may be represented as Equation (Equation 3). The parameters of $U\left(x\right)$ are (*H*, *W*, *a*, *A*, *B*, $\omega_{0}^{2}$, β), where (*H*, *W*, *a*) are the parameters of the WSSR and (*A*, *B*, $\omega_{0}^{2}$, β) are the parameters of the CSSR, all of which have real values higher than 0.
(3)$$U_{2}\left(x\right)=-\frac{1}{2}A\omega_{0}^{2}x^{2}+\frac{1}{4}B\beta x^{4}$$

Further, CSSR is defined as the controlled potential well depth and the controlled potential well width. In the case of the controlled well depth, let *A = B = k*. Then, through analyzing the potential function Equation (Equation 3), two minima, the potential wells, are located at xm=±ω02ω02ββ. The potential barrier is located at *x* = 0, with a height of kω04kω044β4β. So the parameter *k* is independent of the positions of the potential wells and the potential barrier, which are only related to $\omega_{0}^{2}$ and β. For the ease of analysis, we assume that $\omega_{0}^{2}=\beta =1$, Figure 2 shows the effect of different *k* on the potential well depth, where *k* is taken as 0.5, 1, and 1.5, respectively. It is not tough to find that the depth of the two potential wells gradually increases and the walls of the potential wells gradually steepen as *k* increases [39,40,41].

Under the circumstance of the controlled well width, let A=11k2k2, and B=11k4k4. Then, analyzing the potential function Equation (Equation 3), the potential barrier of Equation (Equation 3) exists at *x* = 0, which is similar to the controlled well depth. The difference is that the potential wells are at xm=±kω02ω02ββ, where the depth of potential well is ω04ω044β4β and the width of the potential well is kω02ω02ββ. It is uncomplicated to note that the depth of the potential well is only related to $\omega_{0}^{2}$ and β, and *k* controls the width of the potential well. Consistent with the above, we assume that $\omega_{0}^{2}=\beta =1$, Figure 3 shows the effect of different *k* on the potential well width, where *k* is taken respectively as 0.5, 1, and 1.5. With the increase of *k*, the depth of both potential wells remains constant, the width of the potential well increases, and the bottom of the potential well becomes smoother [42,43,44,45].

Since the potential function of CSwWSSR is generated in the combination of WSSR and CSSR where CSSR has two cases, i.e., the controlled potential well depth and the controlled potential well width, CSwWSSR should also be discussed in two cases. On the basis of the aforementioned analysis of CSSR, we find that $\omega_{0}^{2}$ and β do not affect the potential function either in the controlled well depth or the controlled well width, so we assume that $\omega_{0}^{2}=\beta =1$. Accordingly, the total four parameters controlling CSwWSSR are (*H*, *W*, *a*, *k*) in any case.

The parameters (*H*, *W*, *a*) of the WSSR control the intermediate potential well of the CSwWSSR, which have the same impact on the potential function as for the WSSR, as shown in Figure 4 and Figure 5. In the case of the controlled potential well depth, the depth of the intermediate potential well gradually increases with increasing *H*, the width of the well gradually increases with increasing *W*, and the steepness of the potential well wall gradually becomes slower with increasing *a* in Figure 4a–c. The potential wells on the left and right sides depend on parameter *k* of the CSSR, which can be simply realized that the well depth increases with the increase of *k* in Figure 4d.

For the example of the controlled well width, Figure 5 shows the variation of the CSwWSSR potential function with each parameter. Similar to the controlled well depth, in Figure 5a–c, the depth and width of the intermediate potential well are proportional to the parameters *H* and *W*, as well as the steepness of the potential well walls is inversely proportional to the parameter *a*. Differently, *k* controls the width of the potential wells on both sides, which is positive to the width in Figure 5d. Comparing Figure 4a and Figure 5a, the potential function is exactly the same for both cases when *k* = 1. Surprisingly, it is noteworthy that the CSwWSSR potential function will resemble that of the bistable state. After the analysis of the CSwWSSR, it contains both the characteristic of steep potential well walls in WSSR that allow for stable particle motion and the characteristic of controlled adjustment of the potential wells on both sides in CSSR. More importantly, the CSwWSSR is strongly adjustable, thus it has a significant impact on the SR efficacy and helps the system produce a strong output.

### 2.2. Optimization of CSwWSSR Parameters

In light of the analysis of the CSwWSSR in Section 2.1, We draw the conclusion that the CSwWSSR depends on four parameters (*H*, *W*, *a*, *k*). In order to optimize the output, the particle swarm optimization (PSO) [46,47,48] is used to make the best match between the system parameters and the input signal. Figure 6 provides an illustration of the CSwWSSR model for weak fault detection using PSO. Here, the objective function of the PSO is output SNR from [35], which is defined as follows
(4)SNR=10log10Ad∑i=1NN22Ai−Ad
where $A_{d}$ denotes the amplitude of the fault characteristic frequency in the output power spectrum, ∑i=1NN22Ai−Ad represents the power amplitude sum of noise in the output power spectrum, and *N* stands for the length of the signal. A larger output SNR indicates superior performance in weak signal detection.

Step 1. Preprocess input signal. The adiabatic approximation theory, which stipulates that the driving frequency, signal amplitude, and noise intensity should all be considerably less than 1, was the basis for the initial proposal of SR. However, the majority of real signals are high frequency signals in the presence of severe noise, which does not agree with the theory of adiabatic approximation. A rescaling method was used in [27] to circumvent the restriction that SR is only applied to small parameter signals in an effort to more effectively apply it to numerous fields. The specific method first scales down the high frequency signal to get it into the adiabatic approximation theory’s narrow parameter region, where SR is possible. The compressed signal is then transformed using SR, and finally the output signal is converted back to a high frequency signal with the same scale.

Step 2. Optimize parameters. The PSO process is shown in the dotted box in Figure 6. First, we need to initialize the PSO, including the population size, the maximum generations of iterations, the learning factor, the initial position of particles, the individual optimal position, the population optimal position, etc. Then, the fitness corresponding to the current position is calculated, i.e., the output SNR. In order to prevent the algorithm from entering a local optimum, the step to update the particle positions involves the use of a dynamic inertia factor whose value is correlated with the number of current iterations [49]. Afterwards, the fitness of the updated position is evaluated against the population and individual optimums. The updated position will take the place of the optimal position when the updated fitness is outperforming the individual optimal fitness and the population optimal fitness. The population optimal position is produced after multiple iterations.

Step 3. Calculate the CSwWSSR. The optimal combination of parameters output in the previous step is substituted into CSwWSSR to calculate the output signal. Due to the fact that the SR is based on a phenomenon generated by a nonlinear system, the analytical and approximate analytical methods are no longer applicable. As a consequence, a numerical method is usually used to solve the problem, i.e., fourth-order Runge-Kutta [36]. The solution method is as follows
(5)k1=Δ∗func_CSwWSSRxini,xouti,H,W,a,kk2=Δ∗func_CSwWSSRxini,xouti+k1xouti+k122,H,W,a,kk3=Δ∗func_CSwWSSRxini+1,xouti+k2xouti+k222,H,W,a,kk4=Δ∗func_CSwWSSRxini+1,xouti+k3,H,W,a,kxouti+1=xini+16k1+2k2+2k3+k4
in which $x_{in}\left(i\right)$ is the *i*-th sample value of the input signal after summing the vibration signal and the noise signal, and $x_{out}\left(i\right)$ is the *i*-th sample value of the output signal. Δ is employed to represent the step size, which is scaled from 11fsfs to 11fs∂=fs∂=111fs∂=fs∂=1(fs(fs∂)∂)(fs(fs∂)∂) when using the rescaling method (∂ is scale compression ratio, $f_{s}$ is the frequency of sample, $f_{s\partial }$ is two sampling frequency). func_CSwWSSR() stands for the Langevin equation for CSwWSSR, expressed as follows
(6)$$\left\{\begin{array}{c} func\_CSwWSSR=\frac{dx}{dt}=-\frac{dU\left(x\right)}{dt}+s\left(t\right)+\xi \left(t\right)\\ \frac{dU\left(x\right)}{dt}=\frac{H}{a}\mathrm{sgn}\left(x\right)exp\left(\frac{\left|x\right|-W}{a}\right){\left(1+exp\left(\frac{\left|x\right|-W}{a}\right)\right)}^{-2}-A\omega_{0}^{2}x+B\beta x^{3} \end{array}\right.$$
where s(t) and ξ(t) are vibration signal and noise respectively, (*H*, *W*, *a*, *A*, *B*, $\omega_{0}^{2}$, β) are the parameters of CSwWSSR. We analyzed in Section 2.1 to know that $\omega_{0}^{2}$ and β have no effect on the potential structure and set $\omega_{0}^{2}=\beta =1$. Both *A* and *B* can be represented by *k* in different cases, so the func_CSwWSSR() only contains four parameters (*H*, *W*, *a*, *k*).

Step 4. Diagnose faults. A power spectrum is created from the output signal via the Fourier transform to detect the weak fault and diagnose mechanical faults. In this work, the output SNR and the amplitude of the power spectrum at the fault characteristic frequency are observed and compared to assess the model’s outstanding performance.

## 3. Simulation

### 3.1. Output Analysis of Simulation Signal

To illustrate the effectiveness of the proposed CSwWSSR model, the simulation signal is imported into the CSwWSSR model’s weak signal detection for fault diagnosis. The simulation signal can be represented as
(7)$$x\left(t\right)=A\sin\left(2\pi f_{0}t\right)+\xi \left(t\right)$$
where ξ(t) is additive Gaussian white noise with $\langle \xi \left(t\right)\xi \left(t+\tau \right)\rangle =2D\xi \left(\tau \right)$ where noise intensity *D* = 5 and $\langle •\rangle$ represent statistical average operator. *A* denotes the amplitude of the simulation signal with *A* = 0.25. To be specific, the fault characteristic frequency $f_{0}$ = 140 Hz, the sampling frequency $f_{s}$ = 10,000 Hz, and the data length *N* = 4000. To satisfy the requirement of adiabatic approximation theory to achieve small parameters of SR, the scale compression ratio ∂ = 2000, then the step size Δ=11fsfs∂∂fsfs∂∂=0.2. Figure 7 displays the simulation signal’s time domain waveform and power spectrum. It is obvious that the input signal without SR processing is completely drowned in the surrounding noise at the fault characteristic frequency $f_{0}$ = 140 Hz in the frequency domain diagram, so detecting the fault of simulation signal is challenging. Moreover, the waveform of the signal is not visible in Figure 7a.

Aiming to demonstrate the fault characteristic detection capability of CSwWSSR, three different models are used to process the above simulation signal, that is WSSR, CSSR, and CSwWSSR. In addition, since CSSR and CSwWSSR are divided into controlled well depth and controlled well width, the input signal is subjected to five experiments. For the sake of fairness, the PSO is used for all five experiments to optimize parameters, where the population size and the number of iterations are both set to 50. After the PSO, Table 1 lists the optimized parameter combinations and output SNR, and Figure 8 provides their corresponding output signal time domain diagram and frequency domain diagram.

From Table 1, in view of the output SNR of the above simulation signal, it is seen clearly that the output SNR is: −9.777 dB (CSwWSSR-width) > −9.863 dB (CSwWSSR-depth) > −10.16 dB (WSSR) > −10.85 dB (CSSR-depth) > −11.63 dB (CSSR-width). In Figure 8a, the amplitude of the output signal after WSSR processing at the fault characteristic frequency of 140 Hz is 0.5795. From Figure 8b,c, the amplitude of the output signal after CSSR processing at the fault characteristic frequency is 0.5406 and 0.49. It can be viewed that the amplitude of the output signal at the fault characteristic frequency after CSwWSSR processing is 0.6327 and 0.6292 in Figure 8d,e. The amplitude of fault characteristic frequency is 0.6327 (CSwWSSR-depth) > 0.6292 (CSwWSSR-width) > 0.5795 (WSSR) > 0.5406 (CSSR-depth) > 0.49 (CSSR-width). So the fault characteristic extraction capability is CSwWSSR > WSSR > CSSR. Furthermore, Figure 8b,c show that the fault characteristic frequency is more significantly influenced by the neighboring frequencies, which is evident from the fact that the amplitude of the non-fault characteristic frequency is nearly similar to the fault characteristic frequency after CSSR processing. Based on the analysis hereinbefore, the method of using the CSwWSSR model to implement weak fault detection for simulation signal appeared more suitable.

### 3.2. Capability of Detecting Different Simulation Signals

To evaluate the capability of the proposed CSwWSSR model in detecting different fault signals, it is demonstrated by a series of simulation signals with different fault characteristic frequencies and noise intensities. Consistent with the above, Equation (Equation 7) is used as the input signal by simply changing its fault characteristic frequency $f_{0}$ and noise intensity *D*, in which the fault characteristic frequency $f_{0}$ is from 0.05 to 1500 Hz and the noise intensity *D* is from 1 to 6. In Figure 9a, let the sampling points *N* = 4000, the signal amplitude *A* = 0.25, the sampling frequency $f_{s}$ = 10,000 Hz, the fault characteristic frequency $f_{0}$ = 140 Hz, and the rescaling ratio ∂ = 1000. In Figure 9b, let the sampling points *N* = 4000, the signal amplitude *A* = 0.2, the noise intensity *D* = 2, and the sampling frequency $f_{s}$ = $f_{0}*100$. To meet the adiabatic approximation theory, the rescaling ratio ∂ = $f_{0}*8$. As before, PSO was used to find the best parameter match with a population size and iteration number of 50 and 10, respectively. The average of the five final output SNRs was taken as the output SNR of the current signal. Figure 9 shows the variation of SNR with *D* and $f_{0}$ under three different models.

We can apparently conclude CSwWSSR is preferable to the other two models in terms of its ability to detect fault signals, whether anti-noise or enhancement of high frequency signals in Figure 9. On the one hand, the improvement of CSSR by CSwWSSR is particularly significant, as can be seen that CSSR has no improvement effect on the input signal at low noise intensities, which is manifested by its output SNR being lower than the input SNR at low noise intensities. Although the enhanced performance of WSSR at high noise intensities is noticeably poorer and even worse than the output SNR of CSSR, it still outperforms input SNR in general. With the adequacies of CSSR and WSSR combined, CSwWSSR has an excellent ability to improve for both high and low noise intensities. On the other hand, the curves of the CSwWSSR are always higher than those of the other two models in Figure 9b for signals of various frequencies. Focusing on the above comparison, the suggested CSwWSSR shows more enhancement capability than CSSR and WSSR for the same input signal, different fault characteristic frequencies and noise intensities.

## 4. Experimental Demonstration of Bearing

In the previous section, we demonstrated the effectiveness of the CSwWSSR model for simulation signals. Furthermore, it is applied to practical engineering applications in this section to demonstrate effectiveness of the CSwWSSR model. Here, we adopt the data of rolling bearing provided by Case Western Reserve University, whose sampling frequency is 12 kHz, and speed is 1796 r/min. Taking the outer loop fault as an example, the theoretical value of the outer loop fault frequency is calculated by Equation (Equation 8) which is 107.30 Hz. The parameters needed to calculate the fault frequency are shown in Table 2.
(8)$$f_{i}=\frac{n_{1}N_{1}}{120}\left(1-\frac{d_{1}\cos\beta_{1}}{D_{1}}\right)$$

In Equation (Equation 8), $n_{1}$ represents the number of rolling elements, $N_{1}$ stands for the rotation rate, $d_{1}$ is employed to denote the rolling element diameter, $\beta_{1}$ is the bearing contact angle, and $D_{1}$ represents the pitch diameter. In this way, set the sampling point *N* = 3000 and the noise intensity *D* = 4. Since the fault characteristic frequency does not fulfill the adiabatic approximation theory, using a rescaling method is necessary, with two sampling frequency $f_{s\partial }$ = 6 Hz.

Figure 10 plots the time domain waveform and spectrum of the input signal. It is shown that, with the absence of noise, the spectrum of the input signal is concentrated at 2000–4000 Hz, which resulted in the failure to detect the fault characteristic frequency. Furthermore, the time domain diagram contains waveforms of periodic shocks and vibrations. To accomplish weak fault detection, additive noise is added to the input signal with the purpose of generating the SR phenomenon. As in Section 3.1, we use PSO to achieve the best match between the signal and the parameters. The best matches for the five cases and their corresponding output SNRs are given in Table 3. Comparing the output SNR for the five cases in Table 3, it is clear that output SNR: −8.996 dB (CSwWSSR-width) > −9.702 dB (CSwWSSR-depth) > −9.813 dB (CSSR-depth) > −9.911 dB (CSSR-width) > −11.21 dB (WSSR).

Figure 11 illustrates the time domain and frequency domain diagrams of the output signals for the five cases at the best match. 108 Hz is not hard to obtain from Figure 11, which is close to the fault characteristic frequency 107.30 Hz. In Figure 11a, the amplitude of the output signal after WSSR processing at the fault characteristic frequency of 108 Hz is 0.5423. From Figure 11b,c, the amplitude of the output signal after CSSR processing at the fault characteristic frequency is 0.4721 and 0.4431. It can be viewed that the amplitude of the output signal at the fault characteristic frequency after CSwWSSR processing is 0.5092 and 0.6009 in Figure 11d,e. The relationship between the amplitude at fault characteristic frequency in the five cases is clear: 0.6009 (CSwWSSR-width) > 0.5421 (WSSR) > 0.5092 (CSwWSSR-depth) > 0.4721 (CSSR-depth) > 0.4431 (CSSR-width). Although the fault characteristic frequency amplitude of WSSR is higher than this of CSwWSSR-depth, it is influenced by the low frequency, whose amplitude is almost equal to that of the fault characteristic frequency in Figure 11a. Taking into account, the fault characteristic extraction capability: CSwWSSR > CSSR > WSSR. Unlike the result in Section 3.1, CSSR outperforms WSSR in practical engineering applications. Consistently, CSwWSSR exhibits the excellent capability of weak fault detection. On account of the above analysis. CSwWSSR is also preferable to CSSR and WSSR in terms of effectiveness in practical engineering applications.

## 5. Conclusions

Linear optimal signal processing is typically less effective with a higher SNR in weak signal detection. However, because weak fault detection is regularly hindered by strong noise, the CSwWSSR model we proposed in this paper exploits noise to enhance weak signal detection. In addition to excellent noise utilization, it also exhibits notable features at the fault characteristic frequency, which is one of the benefits of combining WSSR and CSSR. What’s more, by modifying the parameters, it may visibly alter the potential function’s shape directly. The model parameters with the best SNR may be found for different input signals and noise thanks to an optimization algorithm. Then, comparative experiments are conducted for the controlled potential well depth and controlled potential well width to demonstrate the viability and superiority of the CSwWSSR model in fault characteristic detection. The following conclusions are drawn:We analyze the potential functions of WSSR, CSSR, and CSwWSSR to investigate the effects of parameters on the potential structure and dynamical properties. The intermediate potential well of the CSwWSSR is discovered to be controlled by the parameters (*H*, *W*, *a*), which have the same impact on the potential structure as the WSSR, and the potential wells on both sides depend on the parameter *k*. The CSwWSSR parameters are transparent to the control of the potential structure, which means that each parameter controls a different aspect of the potential structure.The output SNR curves of WSSR, CSSR, and CSwWSSR were examined for various noise intensities and various fault characteristic frequencies, demonstrating how well CSwWSSR combines the benefits of WSSR and CSSR both in terms of anti-noise and enhancement of high frequency signals. It has both the features of stable particle motion in WSSR and the controlled adjustment of potential wells on both sides in CSSR.We compare the output signal time domain and frequency domain diagrams of the WSSR, CSSR, and CSwWSSR after PSO optimized parameters in the simulation signal and bearing experiment. In terms of output SNR and amplitude of fault characteristic frequency, it is apparent that CSwWSSR outperforms CSSR and WSSR, which indicates that CSwWSSR has robust engineering applicability in the future.

In future work, we will consider the optimization problem of two-dimensional signals based on multistable stochastic resonance. In the low SNR environment, it will be a new challenge to make full use of SR techniques to overcome the spatial correlation of two-dimensional signals and recover two-dimensional signals disturbed by environmental noise. It is obvious that the results obtained in this study make a good foundation for the discussed future work.

## Figures and Tables

**Figure 1 sensors-23-05062-f001:**
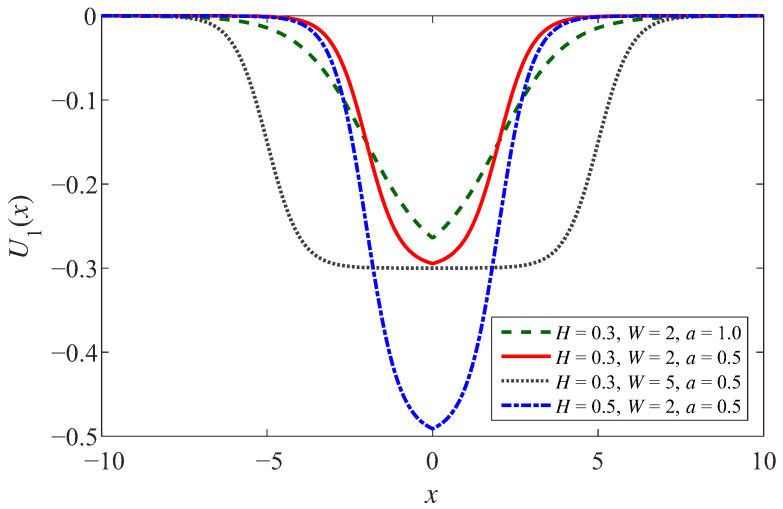
The potential function of WSSR.

**Figure 2 sensors-23-05062-f002:**
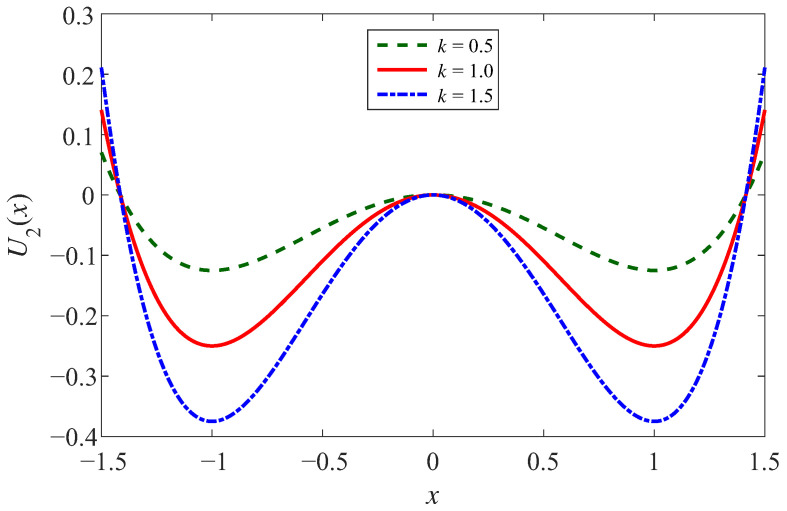
The potential function of CSSR in controlled depth case.

**Figure 3 sensors-23-05062-f003:**
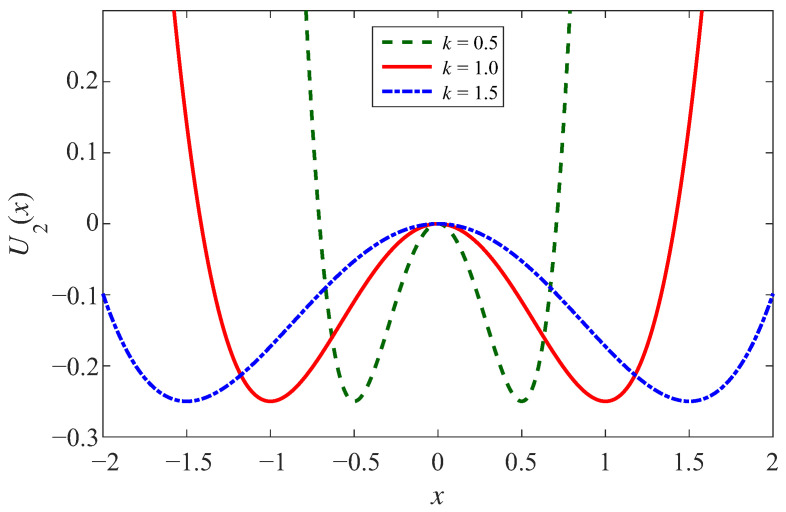
The potential function of CSSR in controlled width case.

**Figure 4 sensors-23-05062-f004:**
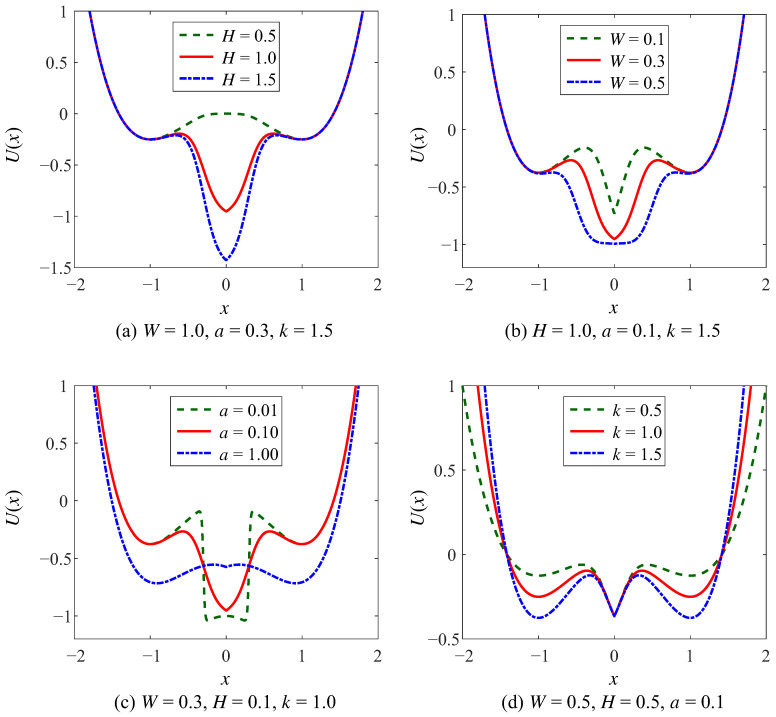
The potential function of CSwWSSR in controlled depth case.

**Figure 5 sensors-23-05062-f005:**
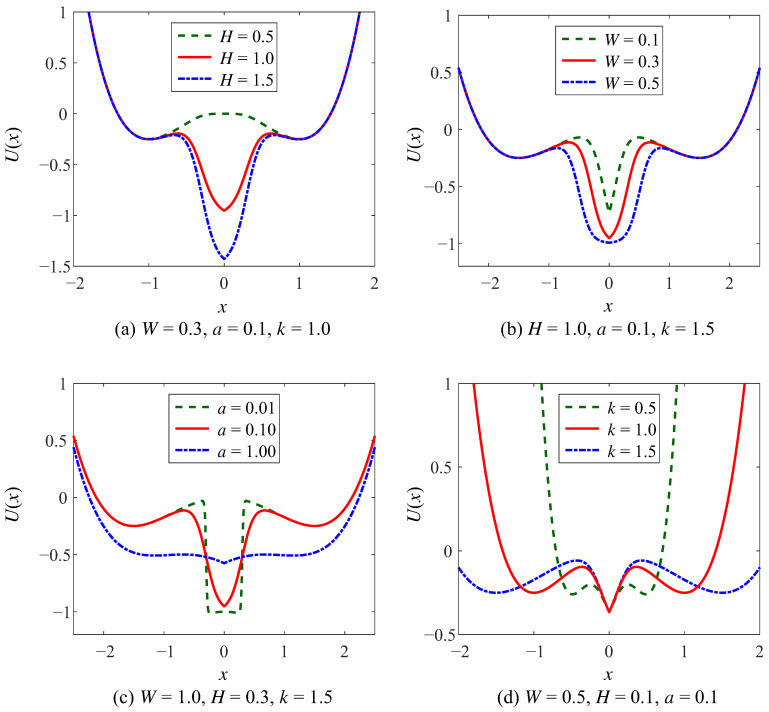
The potential function of CSwWSSR in controlled width case.

**Figure 6 sensors-23-05062-f006:**
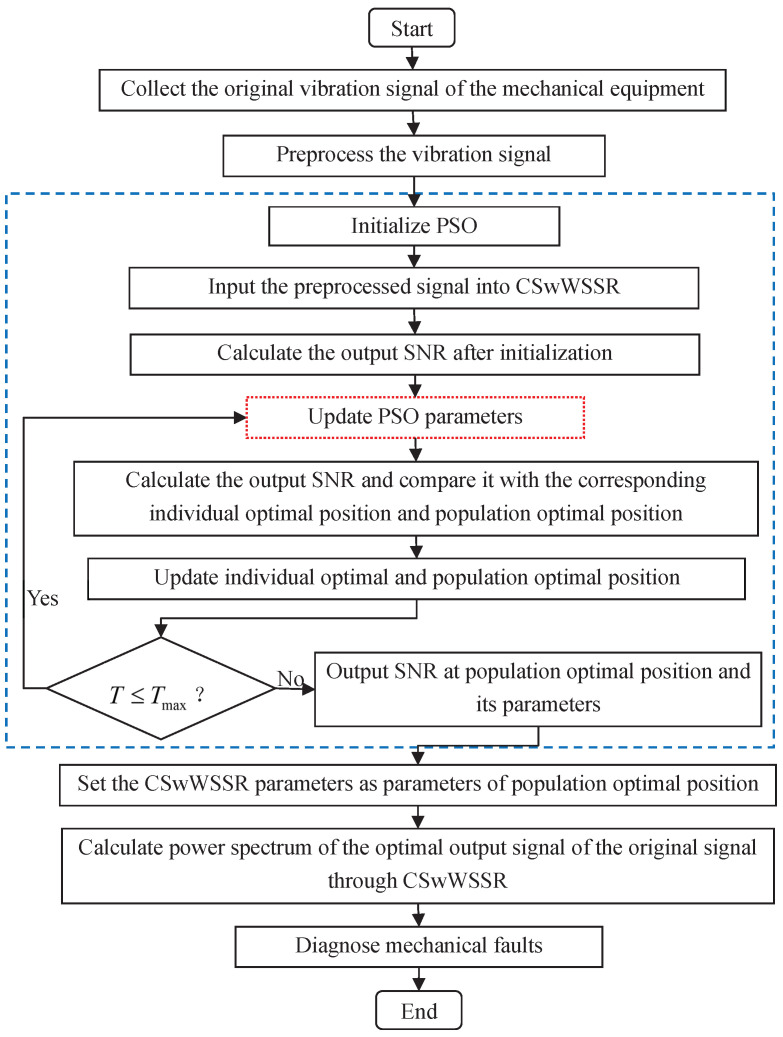
Process flow of CSwWSSR for weak fault detection through PSO.

**Figure 7 sensors-23-05062-f007:**
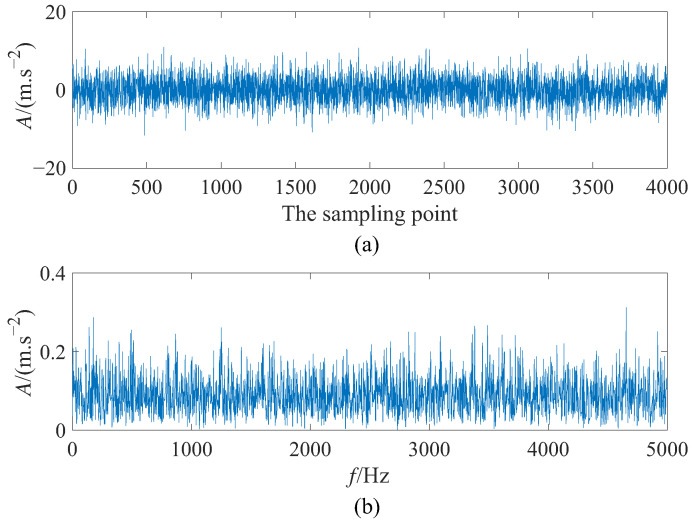
The input signal: (**a**) time domain diagram and (**b**) frequency domain diagram.

**Figure 8 sensors-23-05062-f008:**
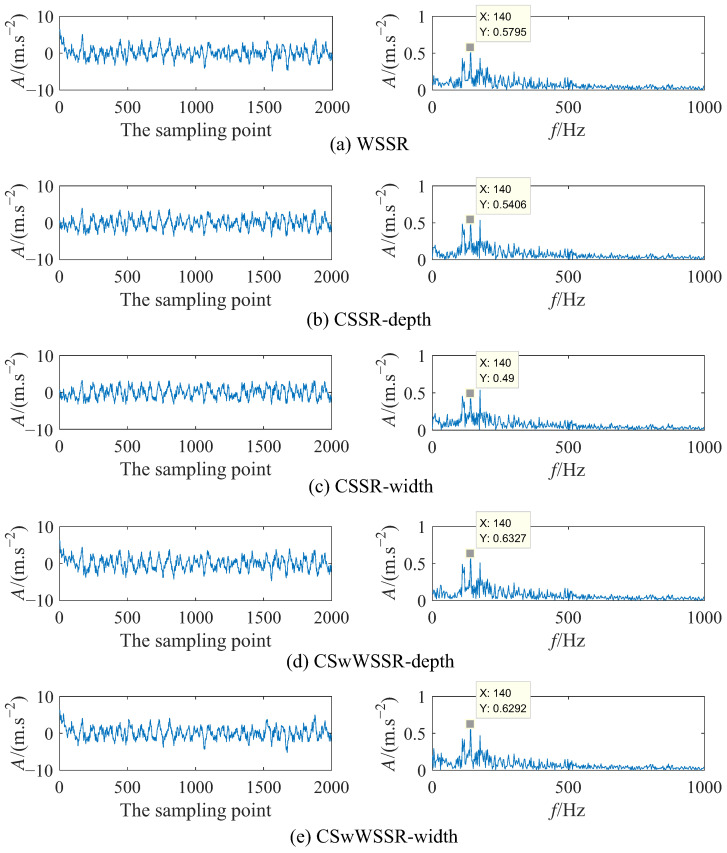
The output signal time-frequency diagram of each model in simulation signal. (**Left** column: Time domain diagram; **Right** column: Frequency domain diagram).

**Figure 9 sensors-23-05062-f009:**
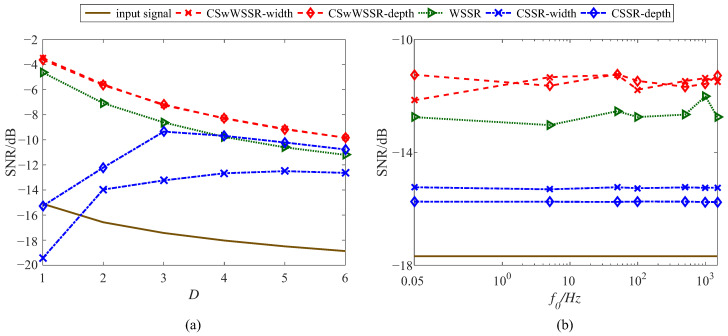
The variation of SNR with (**a**) *D* and (**b**) $f_{0}$.

**Figure 10 sensors-23-05062-f010:**
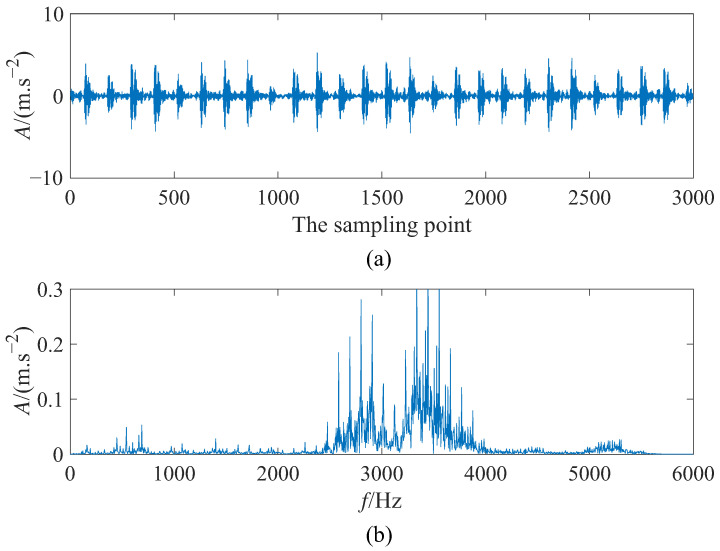
The input signal: (**a**) time domain diagram and (**b**) frequency domain diagram.

**Figure 11 sensors-23-05062-f011:**
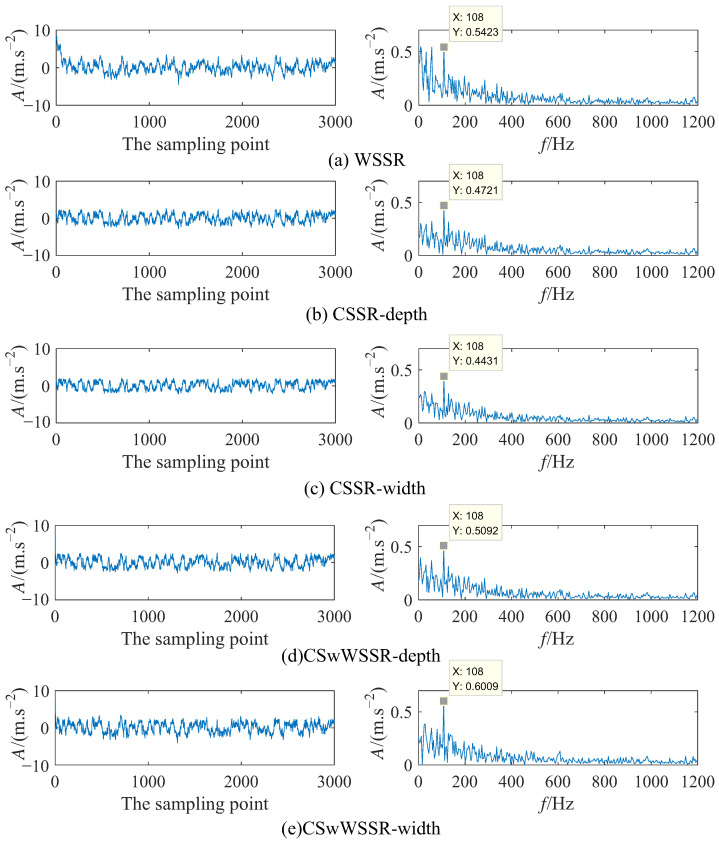
The output signal time-frequency diagram of each model in simulation signal. (**Left** column: Time domain diagram; **Right** column: Frequency domain diagram).

**Table 1 sensors-23-05062-t001:** Parameter combination and output SNR of each model after optimization of simulation signal.

Model	*H*	*W*	*a*	*k*	SNR/dB
WSSR	8.5624	6.2607	3.5634	-	−10.16
CSSR-depth	-	-	-	0.0417	−10.85
CSSR-width	-	-	-	1.6886	−11.63
CSwWSSR-depth	5.5987	5.1328	0.6898	0.0015	−9.863
CSwWSSR-width	1.6607	2.3355	0.1399	3.3592	−9.777

**Table 2 sensors-23-05062-t002:** The main structure parameters of the tested bearing 6205-2RSJEM.

Inner Diameter/mm	Outer Diameter/mm	*D*${}_{1}$/mm	*n* ${}_{1}$	*d*${}_{1}$/mm	Contact Angle/(°)
25.001	51.999	39.040	9.000	7.940	0

**Table 3 sensors-23-05062-t003:** Parameter combination and output SNR of each model after optimization of bearing.

Model	*H*	*W*	*a*	*k*	SNR/dB
WSSR	8.2734	3.8796	0.7568	-	−11.21
CSSR-depth	-	-	-	0.2897	−9.813
CSSR-width	-	-	-	1.1389	−9.911
CSwWSSR-depth	3.4026	3.4026	0.0367	0.1631	−9.702
CSwWSSR-width	3.2067	2.2868	0.0164	1.7698	−8.996

## Data Availability

Not applicable.

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
