# Peer review of "Controlled Symmetry with Woods-Saxon Stochastic Resonance Enabled Weak Fault Detection"

_sensors, 2023, doi:10.3390/s23115062_

Round 1
Reviewer 1 Report
Introduction section to justify the motivation for this study. it is not very clear for a general reader.
- double-check the grammar of the text.
- symbols and parameters in equations should be defined (it is necessary).
- quality of figures should be improved. They are not acceptable for publication.
- reference is not adequate and too short, the authors should cite the following papers in your formulation and introduction.
1-Enhancing active vibration control performances in a smart rotary sandwich thick nanostructure conveying viscous fluid flow by a PD controller
2-Influence of imperfection on the smart control frequency characteristics of a cylindrical sensor-actuator GPLRC cylindrical shell using a proportional-derivative smart controller
Minor editing of English language required
Reviewer 2 Report
- The entire referencelist contains almost exclusively works by Chinese researchers (I found two works by authors from outside the People's Republic of China). This is a strange bias as there are many papers on stochastic resonance and its applications outside of China.
- The authors admit that the CSwWSSR method they propose (by the way - can you find a shorter acronym? Acronyms containing more than four letters are very inconvenient when reading the content) gives better results than the methods known so far; at the same time, the authors present two variants of their method: cntorlled width based and controlled depth based. Which one should be preferred? Is it a universal result or depends on certain factors (which ones?)? Please explain.
- What algorithm do the authors use to determine the Fourier transform? The most common fast Fourier transform algorithm, FFT, is known to perform best when the length of the series is a power of two. Meanwhile, the authors analyze data with a length of 4000, so this algorithm will not work here - hence my question about the transformation method
Reviewer 3 Report
Dear authors, thank you for your manuscript. I enjoyed reading it but some major changes are required but i will suggest some major changes
Point (1). Please change the title “
Point (2). To improve the impact and readership of your manuscript, the authors need to clearly articulate in the Abstract and the Introduction sections the uniqueness or novelty of this article, and why or how it is different from other similar articles. Please remove the terms, sections, and subsections.
Point (3). The Conclusion section is too short. Please expand it by discussing the future directions of your research, especially how it may contribute to your ongoing research about "symmetry".
Point (4). Please substantially expand your review work, and cite more of the journal papers published by MDPI.
Point (5). Some of the references cited are not yet properly formatted. For example, For the references, instead of formatting "by-hand", please kindly consider using the free Zotero software (https://www.zotero.org/), and select "Multidisciplinary Digital Publishing Institute" as the citation format, since there are currently 20 citations in your manuscript, and there may probably be more once you have revised the manuscript. I will suggest following article need to be cite. Some new concepts related to fuzzy fractional calculus for up and down convex fuzzy-number valued functions and inequalities; New Hermite–Hadamard Inequalities for Convex Fuzzy-Number-Valued Mappings via Fuzzy Riemann Integrals; On fuzzy fractional integral operators having exponential kernels and related certain inequalities for exponential trigonometric convex fuzzy-number valued mappings; Hermite-Hadamard inequalities for generalized convex functions in interval-valued calculus; Generalized Preinvex Interval-Valued Functions and Related Hermite–Hadamard Type Inequalities; Higher-order strongly preinvex fuzzy mappings and fuzzy mixed variational-like inequalities; New Hermite-Hadamard type inequalities for -convex fuzzy-interval-valued functions
Best wishes
Round 2
Reviewer 3 Report
Thanks for your nice research.